# Evaluating an equity-focused approach to assess climate resilience and disaster priorities through a community survey

Samantha Lovell[1], Jamie Vickery[1], Paulina López[2], Alberto J. Rodríguez[3], B. J. Cummings[1], Kathleen Moloney[1], Jeffrey Berman[4], Ann Bostrom[5], Tania Busch Isaksen[1], Erika Estrada[6], Cat Hartwell[1], Pamela Kohler[7,8], C. Bradley Kramer[9], Resham Patel[1,9], Amy Helene Schnall[10], Mary Hannah Smith[1], Nicole A. Errett[1,11]*

1 Department of Environmental and Occupational Health Sciences, University of Washington, Seattle, Washington, United States of America, 2 Duwamish River Community Coalition, University of Washington, Seattle, Washington, United States of America, 3 Office of Sustainability & Environment, City of Seattle, Seattle, Washington, United States of America, 4 Department of Civil and Environmental Engineering, University of Washington, Seattle, Washington, United States of America, 5 Evans School of Public Policy and Governance, University of Washington, Seattle, Washington, United States of America, 6 Executive Office of Resiliency and Health Security, Washington State Department of Health, Olympia, Washington, United States of America, 7 Department of Global Health, University of Washington, Seattle, Washington, United States of America, 8 Department of Child, Family, and Population Health Nursing, University of Washington, Seattle, Washington, United States of America, 9 Public Health - Seattle and King County, Seattle, Washington, United States of America, 10 Disaster Epidemiology & Response Team, National Center for Environmental Health, Centers for Disease Control and Prevention, Atlanta, Georgia, United States of America, 11 Department of Health Systems and Population Health, University of Washington, Seattle, Washington, United States of America

* nerrett@uw.edu

**Data Availability Statement:** The de-identified survey dataset will be available via a project published for this study on the the National Science

## Abstract

As the Duwamish Valley community in Seattle, Washington, U.S.A. and other environmental justice communities nationally contend with growing risks from climate change, there have been calls for a more community-centered approach to understanding impacts and priorities to inform resilience planning. To engage community members and identify climate justice and resilience priorities, a partnership of community leaders, government-based practitioners, and academics co-produced a survey instrument and collected data from the community using the Seattle Assessment for Public Health Emergency Response (SASPER), an approach adapted from the Centers for Disease Control and Prevention's Community Assessment for Public Health Emergency Response (CASPER). In addition, we conducted a process and outcome project evaluation using quantitative survey data collected from volunteers and qualitative semi-structured interviews with project team members. In October and November 2022, teams of volunteers from partner organizations collected 162 surveys from households in the Duwamish Valley. Poor air quality, extreme heat, and wildfires were among the highest reported hazards of concern. Most Duwamish Valley households agreed or strongly agreed that their neighborhood has a strong sense of community (64%) and that they have people nearby to call when they need help (69%). Forty-seven percent of households indicated willingness to get involved with resilience planning, and 62% of households said that they would use a Resilience Hub during an emergency. Survey volunteers

Foundation-funded DesignSafe-CI's Data Depot Repository.

**Funding:** SL, JV, PL, AJR, BJ, JB, AB, TBI, PK, RP and NAE were supported by the UW EarthLab Innovation Grants (https://earthlab.uw.edu/grants/). JV, BJC and NAE were supported by the UW Interdisciplinary Center for Exposures, Diseases, Genomics, & Environment (National Institute of Environmental Health Sciences, https://www.niehs.nih.gov/; Award Number P0ES007033). AB and NAE were supported by the Cascadia Coastlines and Peoples Hazards Research Hub (National Science Foundation, https://www.nsf.gov/; Award Number 210713). JV, JB, and NAE were supported by the UW RAPID Facility (National Science Foundation, https://www.nsf.gov/; Award Number 210997). The funders had no role in study design, data collection and analysis, decision to publish, or preparation of the manuscript.

**Competing interests:** The authors have declared that no competing interests exist.

evaluated their participation positively, with over 85% agreeing or strongly agreeing that they learned new skills, were prepared for the survey, and would participate in future assessments. The evaluation interviews underscored that while the SASPER may have demonstrated feasibility in a pre-disaster phase, CASPER may not meet all community/partner needs in the immediate disaster response phase because of its lack of focus on equity and logistical requirements. Future research should focus on identifying less resource intensive data collection approaches that maintain the rigor and reputation of CASPER while enabling a focus on equity.

## Introduction

As the risks from climate change increase across the U.S., community-based organizations (CBOs), government agencies, universities, and other groups are grappling with identifying and implementing the best strategies to protect existing assets, health, and the environment. Despite this, communities that are most impacted by climate change often are not included in efforts to plan, understand, and develop solutions to build resilience to climate risks [1, 2]. While there is broad acknowledgement that public support is needed to advance climate adaptation strategies, engagement is often lacking [3].

Washington State, U.S.A. already experiences the impacts of climate change through heat waves, floods, landslides, coastal erosion, and wildfires, with cascading consequences to human health and wellbeing [4, 5]. For example, the state has seen increasing temperatures, with three of the top ten highest average temperature years since 1895 all happening within the last seven years (2015–2022) [6].

Seattle, Washington State's most populous city, is home to the Duwamish Valley neighborhoods of South Park and Georgetown. In addition to facing high current and projected risk for impacts from climate-sensitive hazards [7], the neighborhoods are mixed residential, industrial, and commercial land use, resulting in substantial environmental hazards. The Duwamish Valley is in close proximity to major transportation hubs including highways, the Port of Seattle, two airports, and multiple flight paths; industrial sites; and the Lower Duwamish Waterway Superfund Site [8, 9]. Flooding is a substantial risk for the City of Seattle, and the Duwamish Valley could see more than six inches of sea level rise by 2050 and nearly two and a half feet by 2100 [4]. Outside of the climate context, there is a need for increased preparedness in the area due to seismic risk, including that presented by a Seattle Fault earthquake, which has the potential to be as high as magnitude 7.5 [5].

The socioeconomic characteristics of Georgetown and South Park also place residents at greater risk for adverse impacts from potential environmental hazards relative to other areas of Seattle. The neighborhoods have a higher percentage of households with racial and ethnic diversity and with incomes below other neighborhoods in the city. As of 2018, nearly one quarter of the approximately 5,600 residents lacked health insurance compared to 13% in the city, and 38% of the population in South Park and 30% of the population in Georgetown are below 200% of the federal poverty level [10, 11]. As a result of disproportionate environmental burdens and socioeconomic factors that increase health risks, there are significant health disparities in the Duwamish Valley [12, 13].

Climate-related priorities can differ based on community attributes and experience. For example, Kreslake (2019) used a survey approach to understand climate-related policy priorities in three communities identified as climate vulnerable and found that residents of different

income levels and races/ethnicities had different perceptions of which types of local mitigation and adaptation measures were most important [14]. Accordingly, intentional, equitable, trauma-informed and community-centered approaches to climate resilience are necessary for both contextualizing and actualizing resilience plans [1]. The deliberate engagement of communities into climate planning has resulted in stronger programmatic and policy design in which local collaborators are more invested in and trusting of the planning and policy design process [15]. In the climate and energy spaces specifically, research has examined different levels and understandings of participation to assess appropriateness of the policies. For example, Radtke (2014 and et al. 2018) describes how bottom-up initiatives in the energy sector result in more engaged and committed community participants as compared to initiatives that are driven top-down [16, 17].

Participatory community-centered research can help demonstrate the disproportionate impacts of climate change on low-income populations, people of color, older adults, people living unhoused, and other groups that have been marginalized [4, 18]. The benefits of community engagement through community-based participatory research (CBPR) have long been recognized in the academic space [19, 20]. The co-production of climate knowledge to inform research and practice is increasingly recognized as an effective, though under-studied, method for engaging community expertise and collection of climate information [21]. Co-production, or co-creation, is "the process of producing usable, or actionable, science through collaboration between scientists and those who use science to make policy and management decisions" [21]. Key to this process is the development and maintenance of strong, trusting relationships between researchers and partners involved, whether they are policymakers or community members [22]. Further, recent studies have demonstrated that integrating communities as partners in research to identify and assess climate health impacts and threats makes the findings more equitable and the output more relevant to communities [23].

The Duwamish Valley has a rich network of longstanding relationships between community members, practitioners, and researchers to support effective community-engaged research. CBOs have worked for decades to advance environmental justice and racial equity in the Duwamish Valley, and the City of Seattle has developed city-community partnerships to engage community members in equity and environmental initiatives [9]. As part of this work, the City has begun development of the Duwamish Valley Resilience District, a collaborative effort to integrate justice and equity in the city's climate adaptation efforts through community engagement and shared decision-making [24].

To help inform and complement the efforts of the Duwamish Valley Resilience District, the Seattle Assessment for Public Health Emergency Response (SASPER) project aims to co-create and pilot a climate adaptation and disaster response needs assessment tool responsive to and informed by the Duwamish Valley community's needs and priorities. The SASPER approach was based on the Centers for Disease Control and Prevention's (CDC's) Community Assessment for Public Health Emergency Response (CASPER) method, a commonly-used rapid needs assessment method that was originally modified by CDC to be undertaken before or following a disaster's impact [25]. Designed to gather household-level information about communities for emergency managers and public health officials through door-to-door surveying, the method has previously been used in pre-disaster contexts to compare household emergency preparedness by housing type [26] and to evaluate the relationship between perceived disaster preparedness and preparedness behaviors [27]. The CASPER method has also been used outside of the disaster context, including to gather information about community health and examine public awareness of various topics [25]. The SASPER approach employed the CASPER method, with several additional steps to ensure the project centered equity and community voice.

Despite an increase in CASPER implementation in the last two decades, publicly available results of the process and summative evaluations of the approach are lacking [28, 29]. A notable exception is an evaluation of a CASPER led by the Washington County Public Health Agency in Oregon, in which the team conducted a hotwash (a structured debrief to obtain feedback from staff) after the exercise, had an evaluator at the site independently evaluate the entire process (including the data collection and hotwash), and compiled the learnings into an After-Action Report [30]. Some important lessons learned from this CASPER included the need to over-recruit volunteers, test and ensure adequacy of language services, and understand the large time and staff commitment for preparing and implementing the assessment [30].

Given the lack of CASPER evaluations, there remains a gap in understanding the strengths and limitations of using CASPER to collect data in different circumstances, including its use as a method for community-engaged applied research. More broadly, current understanding about how community-engaged research approaches can concurrently foster community engagement around climate resilience planning while advancing research on climate health impacts is also limited. In response, this paper briefly presents key findings from the SASPER survey, as well as an evaluation of the SASPER approach, including its modifications to CASPER that sought to center equity and community in the process. This evaluation provides important insights to guide future assessments that seek to engage and center the needs of communities facing environmental injustices in understanding and advancing climate and resilience priorities.

## Materials and methods

### Overview

Building on long-standing community partnerships, the SASPER project is a collaborative effort between the University of Washington, the City of Seattle, the Duwamish River Community Coalition and the organization's Duwamish Valley Youth Corps, the Washington State Department of Health, and Public Health—Seattle & King County. The project aims to assess household- and community-level climate change and health impacts, access to and needs for information and resources to promote resilience, and pathways for community input into ongoing climate change adaptation planning. In addition, it aims to evaluate the CDC's CASPER method, as well as the modifications to this method that sought to center equity and community in the process.

### Community partnership

Our team included members representing the Duwamish Valley community, state and local government, and academic researchers. To promote research reciprocity, improve transparency, balance real or perceived power differentials, and build trust, we grounded our work in a written community partnership agreement [31]. The partnership agreement outlined principles of partnership, decision making, ownership of work, communications, and approaches to community engagement. We also clearly articulated the scope of work of and associated compensation for our primary community partner organization, the Duwamish River Community Coalition. We recognized both our partnership with the Duwamish River Community Coalition and the time of youth volunteers financially to ensure that youth from all economic and cultural backgrounds were able to participate in training and surveying, and provided survey respondents with a gift card to recognize their time, knowledge, and lived experience shared through their survey responses.

## SASPER survey

**Survey and volunteer preparation.** The SASPER survey instrument was developed based on the knowledge of and needs expressed by the Duwamish River Community Coalition, the City of Seattle Duwamish Valley Program, the Washington State Department of Health, and Public Health—Seattle & King County. First, a list of validated or piloted demographic, disaster, and climate questions was compiled from CDC's CASPER toolkit and the National Institute of Environmental Health Sciences Disaster Research Response (DR2) Resources Portal to align with the survey objectives [25, 32]. Next, the project team formed small groups aligned with each survey objective, including community leaders, to prioritize two to three questions each. The questions were then combined into a single, long-form survey, and the full team reviewed and provided recommendations prior to approval and finalization. The survey was then formatted to fit on a double-sided 8.5x11 inch page, per CDC guidance, for use in the field.

To select a representative sample of households to be interviewed, we applied standard CASPER methodology, which includes two stage cluster sampling. In the first stage, 30 clusters (census blocks) were selected with a probability proportional to the number of households within the clusters. In the second stage, interview teams used systematic random sampling to select seven households from each of the selected clusters [15]. Four clusters were selected twice so interview teams selected fourteen households within those clusters.

Leveraging the strength of partner networks, we recruited volunteers for the surveying from the Duwamish Valley Youth Corps, a program of DRCC, the Public Health Reserve Corps, and University of Washington students from various on campus programs, including nursing and public health. The Duwamish Valley Youth Corps is a program of the Duwamish River Community Coalition which provides a broad range of environmental justice and job-related experiential learning opportunities for local youth ages 13–20 from South Park and Georgetown [33]. CDC provided two training sessions in the weeks preceding the survey days to prepare volunteers for door-to-door surveying: a youth-focused training and a separate training for adult volunteers.

Prior to survey implementation, we conducted extensive community engagement through social media, announcements in neighborhood association newsletters, and placement of physical flyers in the selected clusters. During the placement of flyers, it was discovered that one cluster contained no residences. Instead, it appeared that several individuals living unhoused had registered their mailing addresses as the local food bank. This presented an unprecedented opportunity to include those individuals, historically excluded from household-based surveying, in our assessment. We worked with the local food bank manager to recruit the required seven households at the food bank from among those picking up food during our survey days. Participants were asked to self-report whether they received mail at the food bank; however, receipt of mail at the food bank was not used to screen participants (i.e., we included both housed and unhoused individuals in our recruitment efforts).

As the SASPER aimed to engage representative voices and enable equitable involvement, survey materials were translated by a professional translation service into the eight additional languages other than English known to be most commonly spoken in the Duwamish Valley: Amharic, Cantonese, Khmer, Korean, Mandarin, Spanish, Vietnamese, and Somali. In addition, a live phone bank with interpreters in multiple languages was available for surveyors and participants in case participants spoke another language, or were not literate in one of the nine languages that surveys were translated to.

**Survey implementation.** Survey data was collected between October 29 and November 19, 2022. We conducted door-to-door surveying on Saturday October 29, 2022 (approximately

11AM-2PM), Thursday November 3, 2022 (approximately 1PM-6PM), and Saturday November 5, 2022 (approximately 10:30AM-2PM). Where possible (based on number and availability of volunteers), survey teams included two Duwamish Valley Youth Corps members and two adult volunteers each, following best practices for working with local youth. Additionally, as feasible, we paired adult volunteers from different organizations (for example, pairing a university student with a public health professional) to increase age diversity per CDC recommendation, with the potential added benefit of fostering a mutual learning and cross-disciplinary experience. A two-person team also collected responses from unhoused individuals at the food bank on the first Saturday survey day and an additional survey day that occurred on Thursday, November 3rd. Specifically, we recruited a convenience sample of 8 individuals and randomly selected 7 participants for inclusion in the final sample.

Survey participants who completed the survey were recognized for their time with a $25 gift card. The SASPER teams disseminated several educational materials after completing the survey with each household, including a brochure describing the Duwamish Valley Resilience District activities and a handout promoting an upcoming Duwamish River Community Coalition climate justice event, and surveyors completed CDC-developed tracking forms.

Teams made at least two attempts at each selected household before substitution. To further increase representation and provide alternative opportunities for involvement in the survey, after the final survey day the team left flyers with a link to complete the survey online at households surveyors had visited but without succeeding in making contact with anyone in the household.

**Survey analysis.** Paper surveys were manually entered by three researchers (M.S., S.L., and a non-coauthor collaborator) in the Research Electronic Data Capture (REDCap) tools hosted at the University of Washington [34, 35]. A non-coauthor collaborator reentered the surveys and adjudicated any differences between the entries to develop a final dataset. REDCap is a secure, web-based software platform designed to support data capture for research studies, providing 1) an intuitive interface for validated data capture; 2) audit trails for tracking data manipulation and export procedures; 3) automated export procedures for seamless data downloads to common statistical packages; and 4) procedures for data integration and interoperability with external sources.

Data were weighted according to CDC's methods for weighting CASPER survey data, which involves assigning each cluster a weight according to the number of interviews completed in the cluster, the goal number of interviews for the cluster, and the total number of households in the sampling frame [25]. The data were then summarized by survey question.

## Evaluation

**Evaluation approach.** Through a mixed-methods approach, the evaluation design included collection of quantitative data through a feedback survey completed by SASPER volunteers on the first day they participated in door-to-door surveying, and in-depth key informant interviews with project team members to gain deeper insight into the effectiveness, feasibility, and appropriateness of the project.

To guide the evaluation process, we developed a logic model that identifies the main inputs, activities, outputs, and outcomes from the SASPER surveying process (Fig 1). Key inputs to the process include personnel (the project team, Duwamish Valley Youth Corps, and volunteers), the community partnership agreement developed and agreed to by the project organizations, funding, and the CASPER methods from CDC. Activities include the survey development, recruitment and training of volunteers, survey implementation, informing the community about the Duwamish Valley Resilience District and SASPER efforts, and data

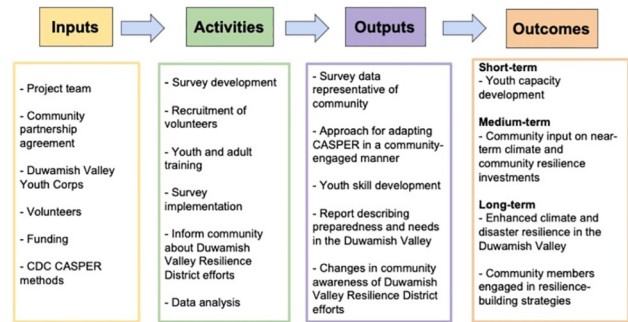

**Fig 1. Logic model documenting SASPER project inputs, activities, outputs and outcomes.**

analysis. Intended outputs of the process are survey data that are representative of the Duwamish Valley community, an approach for adapting CASPER in a community-engaged and -centered manner, Duwamish Valley Youth Corps skills development, a report describing the climate and resilience priorities and needs in the Duwamish Valley, a community fact sheet, community report back events, and changes in community awareness of the Duwamish Valley Resilience District efforts. In the short and medium term, intended outcomes of the process were capacity development of the youth and gathering community feedback on near-term climate and community resilience investments (e.g., City investments in Resilience Hubs, Duwamish Valley Resilience District planning efforts). In the long term, intended outcomes of the SASPER are to enhance climate and disaster resilience in the Duwamish Valley and facilitate a process for stronger engagement of the community in building and implementing resilience strategies. Importantly, however, this evaluation is not able to assess the long-term outcomes of the project.

We drew on culturally responsive evaluation tenets for the assessment. Culturally responsive evaluation presents a framework for ensuring an evaluation process is responsive to context-specific cultural values and beliefs [36, 37]. This involves the integration of cultural values into each step of the process, from design to dissemination, centered with the core principles of responsibility and responsiveness [38]. For example, we shared key evaluation questions with the project leadership for review and feedback. Additionally, a core aim of the evaluation is understanding the degree to which the project was community-engaged and -centered.

**Feedback survey.** A short feedback survey was developed to collect immediate input from adult and youth volunteers on Saturday October 29 and November 5, 2022. The survey provided several statements regarding respondents' level of preparation, skill development, the appropriateness of the surveying method, and willingness to participate in future surveying events. Respondents were asked to rate their level of agreement with these statements on a scale of strongly disagree to strongly agree, with the option to indicate they didn't know the answer. Additionally, a free-response section enabled volunteers to provide information not already captured in the closed-option questions, including suggestions to improve future community disaster needs assessments, ways to improve future volunteer experiences, and other feedback.

**Interviews.** We selected interviewees based on involvement in the SASPER process and included one or more representatives from each of the partner organizations. We sought feedback from additional interviewees within each partner organization to ensure the organizations' perspectives were adequately represented in the evaluation.

We conducted interviews online using Zoom between February 22 and March 7, 2023, and recorded each interview after obtaining verbal consent from the interviewee. After interviews were professionally transcribed, we coded each interview using NVivo Qualitative Analysis software. Two researchers co-coded 20% of the interviews to ensure consistency and reliability. We identified coding discrepancies and adjudicated using consensus based discussion and made refinements to the codebook to clarify code definition for use moving forward.

To analyze the interviews, we used the framework method for qualitative analysis, including deductive and inductive coding strategies [39]. Our first step was drafting a codebook based on our evaluation and interview questions and logic model. We added codes inductively as new themes emerged. After finishing the coding process, we summarized each code by interview and developed a matrix with the summaries to further analyze and synthesize the findings. Finally, we summarized the findings in an analytic memo by code.

### Ethics

The University of Washington's Human Subjects Division (HSD) determined that this research qualified for exempt status (category 2). We submitted an initial application to HSD regarding the door-to-door survey data collection, which was determined exempt on October 25, 2022 (#000165). We received exempt status for a modification to our original submission concerning the addition of the SASPER evaluation (#MOD00015109). Participants in the door-to-door survey were read an overview of study procedures and the intended use of the survey data, and were asked to provide verbal consent to participate after confirming they resided in the home and were 18 years old or older. The evaluation results were not intended to produce generalizable knowledge, but rather to understand what went well and what could be improved from the project. These activities are not considered human subjects research by University of Washington's HSD, and therefore, consent was not required. However, participants in evaluation interviews were asked to provide verbal consent for the interview to be recorded.

## Results

### SASPER survey

We collected 162 surveys from households in the sampling frame: 130 through in-person outreach and an additional 32 through the online survey. One hundred and fifty surveys were completed in English, 10 in Spanish, 1 in Khmer, and 1 in Vietnamese. One survey team used the phone translation service to complete a survey with a resident that spoke Khmer. An additional five surveys were collected though these were excluded as they were from households outside the CASPER clusters. Table 1 presents summarized data by survey question.

The majority of surveyed households reported that at least one member of the household was white (72%). In order of higher to lower percentages, 19% of households reported that at least one member identified as Hispanic/Latino/Latinx, 14% as Asian, 13% as mixed race, and 5% as Black/African American (note that the total exceeds 100% because some households reported having at least one member from more than one race/ethnicity category).

The three issues most frequently cited in the top three concerns for households surveyed were environmental impacts, crime, and cost of living. When asked to rate their level of concern about specific hazards, poor air quality, extreme heat, and wildfires were among the highest reported hazards of concern, with a large percentage of respondents having reported their households experiencing these hazards (59%, 57%, and 43%, respectively).

Most Duwamish Valley households agreed or strongly agreed that their neighborhood has a strong sense of community (64%) and that they have people nearby to call when they need

**Table 1. Responses to SASPER survey questions.** Weighted percentage of households is provided for each response and the 95% confidence interval.

| | Sample N (N = 162) | Weighted N (N = 2,038) | Weighted Percentage* | 95% Confidence Interval |
|---|---|---|---|---|
| **Racial/ethnic composition of household** *(household has at least one member that identifies as the race/ ethnicity)* | | | | |
| American Indian/ Alaska Native | 3 | –** | –** | – |
| Asian | 21 | 277 | 14% | 9%–21% |
| Black/ African American | 7 | 99 | 5% | 2%–9% |
| Hispanic/ Latino/ Latinx | 30 | 378 | 19% | 10%–31% |
| Native Hawaiian or Other Pacific Islander | 4 | –** | –** | – |
| White | 117 | 1,474 | 72% | 62%–81% |
| Mixed | 19 | 263 | 13% | 8%–20% |
| **Age composition of household** *(household has at least one member in the age bracket)* | | | | |
| Less than 2 years old | 9 | 135 | 7% | 3%–14% |
| 2–17 years old | 32 | 413 | 20% | 14%–29% |
| 18–64 years old | 113 | 1,414 | 87% | 82%–91% |
| 65+ years old | 18 | 229 | 14% | 9%–22% |
| **What are the top three issues of concern for your household?** | | | | |
| Environmental impacts | 100 | 1,263 | 62% | 53%–70% |
| Crime | 77 | 990 | 49% | 42%–55% |
| Cost of living | 73 | 925 | 45% | 38%–53% |
| Housing affordability | 56 | 729 | 36% | 28%–45% |
| Racial and ethnic inequality | 41 | 510 | 25% | 20%–31% |
| Civil unrest | 28 | 325 | 16% | 11%–22% |
| Healthcare access | 26 | 337 | 17% | 11%–24% |
| Food security | 26 | 299 | 15% | 10%–21% |
| COVID-19 | 18 | 221 | 11% | 7%–17% |
| Duwamish Superfund site | 14 | 192 | 9% | 5%–17% |
| **Hazards rated of "high concern" by households** | | | | |
| Poor air quality | 110 | 1,421 | 72% | 62%–80% |
| Extreme heat | 82 | 1,052 | 53% | 44%–61% |
| Wildfires | 72 | 925 | 46% | 36%–57% |
| Flooding from heavy rains and/or sea level rise | 42 | 530 | 27% | 21%–33% |
| Extreme cold weather or severe winter storms | 31 | 380 | 19% | 14%–25% |
| Sewage overflow during rain incidents | 26 | 287 | 15% | 9%–23% |
| Contaminated local food sources | 25 | 283 | 14% | 10%–21% |
| Earthquakes | 19 | 221 | 11% | 7%–17% |
| Droughts or water shortages | 18 | 205 | 10% | 6%–18% |
| **Does anyone in your household have a health condition that you think could be worsened in a disaster or an environmental hazard?** | | | | |
| Yes | 72 | 915 | 46% | 37%–55% |
| No | 79 | 996 | 50% | 41%–59% |
| **Do you or someone in your household require medical equipment or supplies that require electricity?** | | | | |
| Yes | 16 | 205 | 10% | 7%–16% |
| No | 138 | 1,733 | 87% | 81%–92% |
| **What are your top 3 information sources about disasters/ environmental hazards?** | | | | |
| Internet | 105 | 1,329 | 65% | 56%–74% |
| Social media | 83 | 1,047 | 51% | 42%–61% |
| Friends/family/word of mouth | 70 | 863 | 42% | 33%–53% |
| TV | 43 | 597 | 29% | 21%–40% |

*(Continued)*

**Table 1.** (Continued)

| | Sample N (N = 162) | Weighted N (N = 2,038) | Weighted Percentage* | 95% Confidence Interval |
|---|---|---|---|---|
| Text message/alert | 47 | 558 | 27% | 20%–36% |
| Cell phone | 44 | 518 | 25% | 19%–34% |
| Radio | 35 | 456 | 22% | 16%–30% |
| Newspaper | 27 | 333 | 16% | 11%–23% |
| Community health clinic | 10 | 117 | 6% | 3%–12% |
| **Which three actions should be prioritized to increase climate and community resilience in your community?** | | | | |
| Green infrastructure | 122 | 1,547 | 76% | 68%–83% |
| Support community-centered/led projects | 82 | 1,090 | 54% | 45%–62% |
| Improved stormwater management | 76 | 941 | 46% | 39%–54% |
| Flood protection | 70 | 850 | 42% | 34%–50% |
| Improved transit | 65 | 825 | 41% | 33%–48% |
| **Would you or any member of your household like to be involved in Duwamish Valley Resilience District work?** | | | | |
| Yes | 76 | 932 | 47% | 35%–58% |
| No | 44 | 548 | 28% | 20%–37% |
| **How would you or a member of your household like to be involved in the Duwamish Valley Resilience District work?** | | | | |
| Receive information and provide feedback through VIRTUAL community forums / meetings | 61 | 741 | 36% | 27%–46% |
| Receive information and provide feedback through IN-PERSON community forums / meetings | 40 | 490 | 24% | 17%–33% |
| Be part of a community advisory group that meets regularly and shapes the formation of the Duwamish Valley Resilience District | 31 | 380 | 19% | 13%–25% |
| **Would you or any member of your household use one of these Resilience Hubs during an emergency?** | | | | |
| Yes | 98 | 1,230 | 62% | 50%–72% |
| No | 21 | 261 | 13% | 8%–20% |

*Percentages have been rounded to the nearest percent

**Response options with 5 or fewer responses have not been weighted and are not provided.

help (69%). Approximately half (46%) of surveyed households reported that at least one member of the household has a health condition that could be worsened in a disaster or environmental hazard and 10% reported that someone in the household relies on medical equipment that requires electricity.

When asked which actions should be prioritized to increase climate and community resilience in their community, households identified green infrastructure (for example, permeable pavements, tree planting, and other infrastructure that manages water and/or reduces heat), community-centered/led projects, and improved stormwater management as top actions.

The survey provided information about the City of Seattle's Duwamish Valley Resilience District work, including the city's plans to establish Resilience Hubs in the Duwamish Valley. Resilience Hubs are publicly- and community-owned buildings that can provide services during emergencies. Forty-seven percent (47%) of households indicated willingness to get involved with Duwamish Valley Resilience District work and 62% of households said that they would use a Resilience Hub during an emergency.

## SASPER evaluation

The purpose of the evaluation is to assess the appropriateness, feasibility, and effectiveness of the SASPER approach to meet partner needs and goals and to provide recommendations for

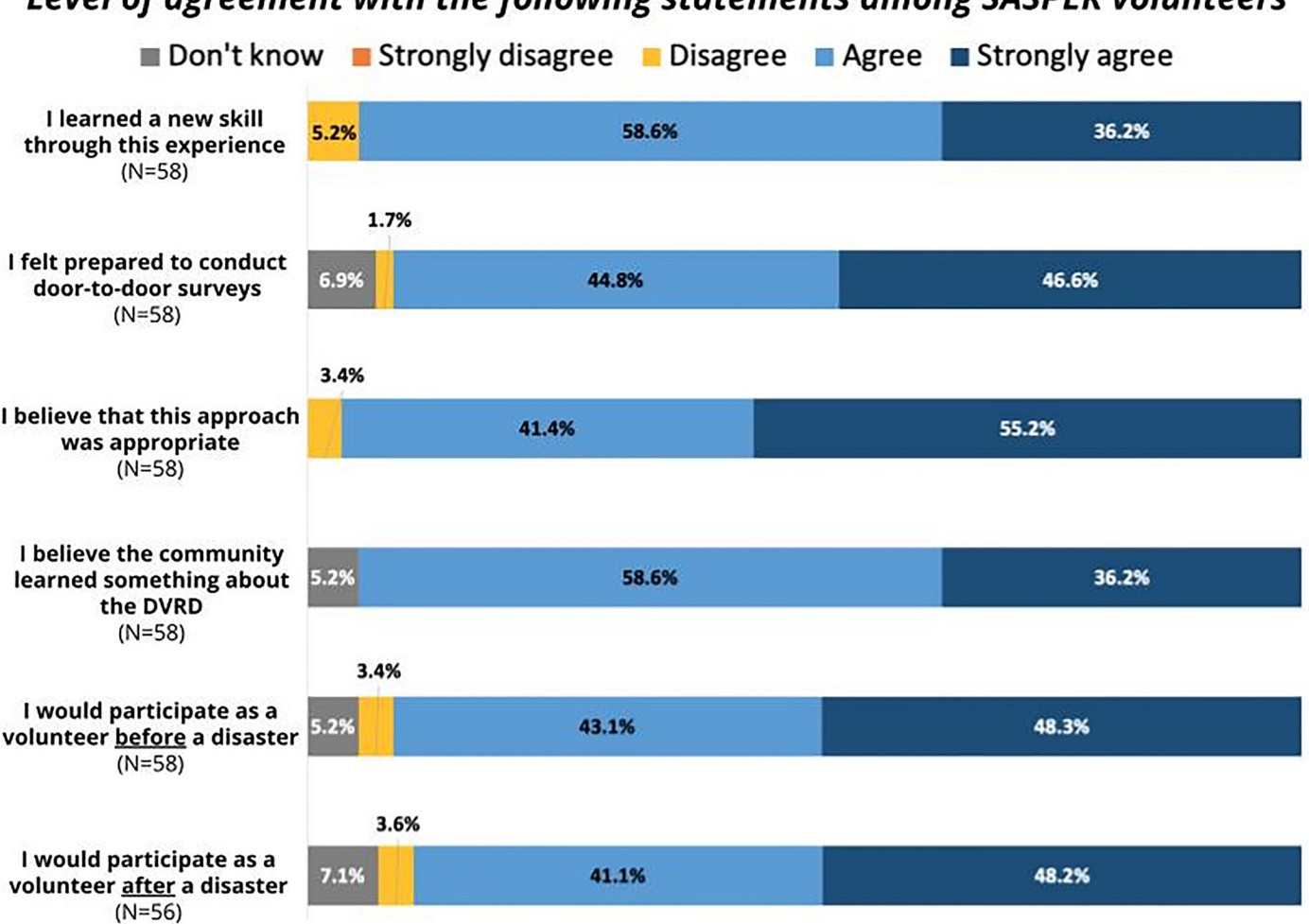

**Fig 2. Adult and youth volunteer responses to day-of feedback survey.**

future assessments and research. We present an overview of results from the volunteer feedback surveys and takeaways from the interviews, including as related to survey preparation and implementation, survey outcomes, and lessons learned.

**Feedback survey.** Fifty-eight volunteers participated in the feedback survey. The vast majority of volunteers who participated in the feedback survey, regardless of whether they were youth or adults, agreed or strongly agreed with each statement provided in the feedback survey (Fig 2). Specifically, 96% of respondents strongly agreed or agreed that the approach was appropriate for collecting information representative of the community, 95% that they learned a new skill and that the community learned something about the Duwamish Valley Resilience District work through their outreach, 92% that they were prepared to conduct door-to-door surveys, 91% that they would participate in future assessment before a disaster, and 87% that they would participate in a future assessment after a disaster.

**Interviews.** Eleven interviews were conducted with project team leadership on Zoom, with a range of 20 minutes to one hour per interview. Five interviewees were affiliated with University of Washington, four with government agencies, and two with CBOs. Interviewees

provided feedback on the survey preparation and implementation process, survey outcomes, and broader lessons learned from the survey. Interviewees shared their perceptions on the process for developing the survey, implementing the survey, and recommendations for improving these processes based on challenges experienced, as well as survey outcomes (i.e., representativeness of the information collected and the impact of the experience on volunteers, with a focus on the Duwamish Valley Youth Corps members).

## Formative evaluation findings

**Survey development.**   The project team's approach to survey development, using small groups to prioritize topic-specific questions (see Methods), was described as efficient and fair. One interviewee expressed that it,

> "was pretty well done. . .it was really well communicated. . .And with people's limited time and capacity, I think having had it staffed by people who could do a lot of the lifting and then nudge people and schedule meetings and stuff, I think it was handled extremely well, given the time pressures,"

(Govt. 4)

Despite this streamlined process, it was still difficult to reduce the number of questions on the survey to ensure the content fit on one page (double-sided) and could be completed within 15–20 minutes, as recommended by CDC, with one interviewee noting, "I think it [the survey] was useful. I think the first draft that was a lot more comprehensive would have been more useful, but I do understand," (Govt. 3). Additionally, the survey development took longer than anticipated, truncating the time available for local language speakers to review and provide feedback on the survey and translations. This limited the accessibility of the survey, particularly for individuals who do not speak English, as one interviewee said, "Well, working with the public, obviously, some will understand it and some will not. I think it's always challenging for people that are non-English speakers just because of the technical wording of things," (CBO 2).

**Survey implementation.**   Interviewees highlighted the importance of proactive modifications to the standard CASPER that were made to center equity and community including translating survey materials into the most common languages in the Duwamish Valley, providing a live interpretation line, compensating households with a gift card, and engaging youth as volunteers. Nearly all the interviewees described how involving the youth in the SASPER was critical to its success and provided an important capacity building opportunity:

> "having the Duwamish Valley Youth Corps participate was incredible for a variety of reasons in terms of what the spirit of it was and engaging the young people. . .But [also] willingness to participate. . .folks noted that the minute the youth corps member had to leave towards the end of the day, the willingness for folks to. . .engage in conversation also plummeted,"

(Govt. 1).

Reactive adaptations to the original project plan were also made in response to logistical challenges to achieve the necessary number of surveys. These changes included adding an extra survey day and providing an online option for households to complete the survey. Despite these additional efforts, the final survey number was slightly short of the CDC-recommended number for statistical significance (162 included in SASPER survey sampling frame

**Table 2. Challenges and recommended solutions specific to survey preparation and implementation.**

| Survey Process Stage | Challenge | Recommendation |
|---|---|---|
| **Development** | Refining survey to the final, short length | Use small groups focused on specific topics to identify questions |
| | | Clarify the length limitations of the final survey to the project team |
| | | Share examples of previous CASPER questionnaires with the project team |
| | Identifying question response options that are relevant to the community | Consider community focus groups to support survey development and identify contextually-relevant questions and responses |
| | Finalizing survey and translations in required time | Build out clear timeline, particularly if working with multiple partners |
| **Content** | Ensuring terminology is accessible and appropriate for all community members | Do not use jargon or acronyms |
| | | Ensure local language speakers review and provide feedback on survey translations |
| | | Use supplementary content to provide concrete examples (e.g., of communication materials) for survey respondents |
| | Communicating questions with multiple response options | Do not use long lists; include short list of relevant options or open response |
| **Training** | Providing audience-relevant and appropriate training for all volunteers | Hold separate trainings targeted for each volunteer type (i.e., if including youth volunteers) |
| | | If working with youth volunteers, ensure training is engaging and interactive |
| | Implementing training within time constraints | Develop a robust schedule for training which allows time for survey practice and team-building |
| **Notice & Outreach** | Raising awareness of surveying efforts among community | Increase outreach efforts: mailers once the survey clusters are identified, radio, etc. |
| **Implementation** | Conducting adequate number of surveys according to CASPER methods | Use a range of times for surveying (mix of weekend/ weekdays and afternoon/ evening) |
| | | Conduct surveying in season with adequate daylight for evening surveying |
| | | Consider including online option for the survey from the start, with multiple language options |
| | Ensuring enough volunteers for necessary number of teams | Overrecruit volunteers (particularly adult volunteers if working with youth) |

compared to 168 minimum required for the CASPER). As such, several interviewees suggested future surveying efforts include an online option from the start of the surveying process and to add survey days.

**Summary of process challenges and recommendations.** Interviewees, overall, had favorable comments about the survey preparation and implementation process, though nearly every interviewee noted constraints to and challenges of the process. Challenges and recommendations described by interviewees are summarized in Table 2.

## Summative evaluation findings

**Representativeness.** Interviewees described the representativeness of the SASPER survey respondents in different ways. Specifically, most interviewees noted that the racial/ethnic breakdown of the survey respondents did not appear to be representative of the Duwamish Valley community, and one interviewee described that while the survey results cannot be directly compared to census data (because the survey captured only household representation of particular races and ethnicities, not individuals who identified as a particular race or ethnicity), the results did appear to overrepresent households with at least one White member and underrepresent households with at least one Black, Indigenous, People of Color (BIPOC) member. However, several interviewees described that the content of the responses did appear to align with the priorities of community members according to the interviewees' perceptions and previous work.

Several interviewees described how the surveying method itself was not representative because it did not take deliberate steps to center the voices of BIPOC individuals. As such, one interviewee expressed, "I really never understood and or agreed with the randomized [approach] because I feel like some. . .important places in the neighborhood were missed" (CBO 1). Several interviewees expressed that other types of door-to-door surveying may be more beneficial in specific circumstances, such as specifically surveying BIPOC individuals, using community service providers to survey specific racial/ethnic groups, and surveying all households in a given area of interest. On the other hand, an interviewee described how leveraging an accepted approach contributed to the value and impact of the information within their organization: "I now have. . .input that I received through a statistically sound process. . .based on the CDC on what are community priorities. . . I can go back to my people and say this is it, this is where we need to put money, this is what we need to prioritize in terms of work plans," (Govt. 3).

Interviewees also described limitations of the CASPER approach to make comparisons within the CASPER sampling frame, as community partners were interested in making comparisons between the two neighborhoods (South Park and Georgetown) that comprise the Duwamish Valley. While one option would have been to conduct two CASPERs (one in each neighborhood), this would have been challenging given the small populations that comprise each neighborhood, and the logistical and resource needs of conducting two CASPERs.

**Adult volunteer experience.** The majority of interviewees able to speak to the volunteer experience described how it was a unique and important experience for adult and youth volunteers, one saying the learning went "beyond hard skills" (Govt. 4). For adults, in addition to building survey skills and knowledge about the CASPER process, the experience enabled a greater understanding and awareness of the Duwamish Valley community, particularly for those who have never been to the area, and the importance of community-engaged research efforts. In voicing this, one interviewee said, "having this very deeply important experience of going into the community, talking with people, it highlights the importance of that as a way to understand the needs—short and long-term—and experiences, concerns of the people who really should be driving. . .our work and solutions," (UW 3).

**Duwamish Valley Youth Corps volunteer experience.** Interviewees with insight into the experiences of the youth volunteers described how they had developed new interpersonal and public speaking skills, learned about public health and research methods, and built confidence by being part of the broader project effort related to community priorities. One interviewee noted, "I also think it was great to be able for them to feel like they are participating in this important effort, that they understand their power that they have by doing this" (CBO 1). However, interviewees noted that the surveying process could have been tailored more to improve the youth experience. Specifically, one SASPER interviewee noted that, during the portion of the surveying that the adults led, the youth could have been conducting another form of data collection or doing something more productive than waiting for their turn to ask specific survey questions.

## Lessons learned for future survey efforts

The SASPER provided an opportunity to test the CASPER approach with modifications, and several interviewees questioned when using an adapted CASPER approach is appropriate. Key limitations of the approach identified by the project team include that it is extremely resource and labor intensive, it does not enable a centering of BIPOC voices, and it provides a limited ability to make demographic and/ or geographic comparisons. Interviewees described the unique nature of the SASPER project in terms of the number of project partners representing

different sectors and how valuable the process itself was, in addition to the information gained through the process. Though the project was rooted in the CASPER method, there were many modifications built in to center equity and community in the process, and the team also had to be flexible and develop creative solutions to overcome new challenges. One interviewee commented on the novelty of the SASPER: "I have not been aware of any other project like this happening [at UW]. And pick a thread. . .a door-to-door-based surveying project, a climate change-resilience project, a community-involved, community-led, community-driven, community-collaborated project, a project working with youth. . ..And I wish there was more of it, or hope that there'll be more of it," (UW 3).

**Feasibility.**   Interviewees had mixed responses when considering the feasibility of conducting a similar surveying effort, though most expressed that it would be feasible, particularly prior to a disaster, with several caveats. The project was resource- and labor- intensive, and extremely challenging to coordinate logistically. However, as the team has now been through the process once and has developed necessary skills, several interviewees expressed that it would be easier to do again. The caveat is that replicating the surveying with the same adaptations to center community and equity requires significant resources. One interviewee recalled, "going back to the point CDC made early on about you could do a CASPER in a week. Well, you can't really do that if you're really trying to get the community buy-in and making sure everyone's at the table. And so. . . if you're thinking about doing this as a community-engaged process for planning or otherwise, to make sure that you have built in enough time and financial resources to do that from an equitable perspective," (UW 1).

**Co-production.**   In describing the partnership among the organizations, all interviewees able to speak to it had positive comments about the collaboration, though several academic and government partners said they would defer to the community partners in making that judgment. In providing the community-partner perspective, an interviewee conveyed that, "I do feel like we gave good information about the community to be represented and also recommendations. . .So I don't know if it was one hundred percent co-design, but it was definitely a good share [of] responsibility," (CBO 1). Several interviewees described elements that were important to building the project collaboratively, including having a CBO at the table, developing the community partnership agreement, UW taking on logistical work to alleviate the burden on community partners, having staff with trusted relationships to function as "connectors" between the partner organizations, and altering project plans as needed in service of community priorities. An interviewee noted that the partnership, itself, has had broader impacts across the community: the project has, "set a precedent on how partnerships and collaboration between academia and the UW [University of Washington] community happen in the Duwamish Valley and I don't think we all, including community partners, will expect any less. So I think that's very powerful," (Govt. 3).

## Discussion

This study provides important new insights regarding the challenges and opportunities of using a CASPER approach in community-centered research concerned with equity. The SASPER survey highlighted that Duwamish Valley community members are highly concerned about environmental hazards, the impacts of which are being worsened by climate change; there is a strong sense of community and connectedness in the Duwamish Valley; and community members are interested in being involved in resilience building efforts. The key learnings of the evaluation for future research and practice are 1) the SASPER innovations were valuable but resulted in a more resource-intensive and logistically-challenging project and 2) while the SASPER may have demonstrated feasibility in a pre-disaster phase, the CASPER

method may not be appropriate to use post-disaster in meeting the needs and priorities of partners due to its lack of focus on equity and other logistical limitations.

The SASPER survey underscored that, absent climate change, those living in the environmental justice community of the Duwamish Valley are concerned about their ongoing exposure to pollution. Furthermore, environmental and climate hazards, including air pollution, extreme heat, and flooding, are high priorities for community members. These findings are consistent with and provide validation of previous, more informal community surveys, pollution mapping, and visioning results dating back to 2007. Air pollution, in particular, has long been identified as a leading health concern for Duwamish Valley residents [10]. The survey results also highlight the community connectedness and capacity in the neighborhoods, which are critical elements of efforts to build communities more resilient to the impacts of climate change. Finally, the survey showed strong support for resilience building efforts in the Duwamish Valley and the community's interest in being involved in these efforts. These findings reflect input gathered during development of the City of Seattle's Equity and Environment Agenda and Duwamish Valley Action Plan, during which one community member stated, "[We need] community controlled climate resiliency planning. We believe in the self-determination of our communities and already have visions and plans for what makes us resilient and healthy" [10, 40]. Accordingly, these survey results have and will continue to inform efforts by the project partners, including the City of Seattle and the Duwamish River Community Coalition's initiatives to build resilience in the Duwamish Valley.

The SASPER evaluation highlighted the importance and impact of engaging the Duwamish Valley Youth Corps in the project. This involved hosting a youth-focused training to prepare the youth corps members for surveying, implementing survey practice days for youth, and including two youth in as many of the approximately 15 survey teams as possible. To the best of our knowledge, youth have not been involved in other CASPER implementations, though there is some documentation of youth engagement in community assessments and, more broadly, in community-health research and community-engaged projects. A study from nearly 30 years ago supported routine engagement of youth in community health needs assessments by presenting examples of involvement and highlighting the benefits [41]. Specifically, Israel and Ilvento described how a Florida county health agency partnered with a local high school to train and guide students in developing, implementing, and analyzing a community survey, with students expressing their desire to continue community-involved work in the future [41]. Despite this, a 2017 study noted the dearth of youth involvement in community assessments [42]. Focusing on Massachusetts, Chen found that 20% of community health needs assessments in the state engaged youth, though the engagement was mainly as research subjects, not as participants in the process [42].

There are several benefits of engaging youth in research and community projects, both for the youth participants themselves and for the projects. Specifically, youth-engaged research can make the research more relevant by incorporating the unique perspectives of youth [43, 44]. CBPR involvement and civic engagement activities can enable youth to develop new skills and have exposure to new opportunities and perspectives on community issues and solutions —potentially leading to more civic engagement in adulthood [44–46]. Indeed, SASPER adult volunteers and project leadership highlighted how critical youth team members were for engaging community members and conducting the surveys. Furthermore, the day-of feedback surveys indicated that the youth involved in the SASPER felt they built new skills and were interested in participating in future efforts.

As such, we recommend working with youth for similar community projects, though research teams and partners must be prepared and plan for the challenges of and considerations for youth engagement. Most importantly, funding, materials, training, and processes

should be tailored to the needs and skills of youth [42], and it may be beneficial to have a youth program leader take an active role in planning and implementing training. Additionally, project leaders should consider training adaptations and potential adaptations to the project scope to increase the youth skill development opportunity of the research. Finally, project leaders must ensure that academic and institutional policies for youth engagement and protection are followed, and these added steps should be built into the planning timeline.

Our evaluation highlighted several limitations of the CASPER approach for pre-disaster assessments, and interviewees noted that it may not always be the appropriate or preferred approach given its lack of focus on equity, its rigidity in design, and the resources required to conduct a CASPER. First, the CASPER method was not developed as an equity-centered tool: it is a rapid needs assessment designed to obtain data representative of a population. One of the main benefits of the approach is that it has been validated as providing population-representative data [47, 48]. Compared to convenience sampling, CASPER's two-stage cluster sampling strategy is less biased. However, compared to simple random sampling, the data needed to implement it is often more-easily available (i.e. identifying clusters using census data), it requires a smaller sample size to be representative than simple random sampling, and the cluster sampling approach results in significant time savings and easier implementation (e.g., because teams need to only go to one or two areas and spend less time driving across town, finding households, etc.) [49]. As a result, data from CASPERs are used and accepted widely by public health and emergency preparedness managers for official purposes, including for community health assessments [49–52]. This was noted by SASPER evaluation interviewees as an important benefit of the approach: the perception of validity and official nature of using a CDC method makes it more likely that audiences, such as government agencies, will accept the data and use it to inform resourcing and policy decisions.

Despite these positive elements, the nature of the sampling does not enable focusing on specific racial/ethnic or language-speaking groups, and lacks sufficient power to do comparisons in most situations. As stated in the CASPER toolkit, the method is not appropriate to assess specific populations, and, if that is the goal, researchers should use another approach [25]. This is a notable limitation for community assessments that use CASPER as a data collection method, as health agencies and hospitals often use such studies to understand the needs of populations that are at-risk of experiencing a higher burden of negative impacts from environmental hazards and to make decisions regarding resource allocation and supportive programming [53]. The limitations identified by our partners echo those from a prior CASPER used for a comprehensive Community Health Needs Assessment in North Carolina, which noted that challenge associated with the inability to describe health disparities across different demographic characteristics [51].

The other identified disadvantage of the CASPER approach is that it is highly resource-intensive, particularly when additional adaptations are made to engage community members and groups equitably. While CDC states that CASPER is "quick, relatively inexpensive, [and] flexible," this was not our experience. The project was supported by a $75,000 grant, which only covered a small portion of the staff time, supplies, and compensation to youth volunteers, survey respondents, and community partner organizations needed for survey development, data collection and analysis, and the dissemination of survey findings. Those grant funds were supplemented by resources provided by the University of Washington and external partner organizations, including translation services, food, additional student, staff, and faculty time, and communications resources. University of Washington faculty and staff, as well as staff from the Washington State Department of Health, Public Health—Seattle & King County, the Duwamish River Community Coalition, and the City of Seattle Duwamish Valley Program participated in months of meetings to develop and refine the survey. The administration of the

survey was also extremely challenging to coordinate logistically, requiring the efforts of 45 adult and 27 youth volunteers, who cumulatively spent over 470 hours collecting door-to-door surveys across the three in-person data collection days. A previous evaluation also emphasized the personnel, financial, and logistical challenges of the approach [25, 30]. The resource demands of the CASPER approach may be a barrier to using this method to collect data for community-engaged pre-disaster assessments. This will limit the replicability and scalability of our modified CASPER approach as a tool for community-engaged research in other contexts, especially if the project does not include a partner like a university or other entity that is able to take on the associated logistical and financial burdens.

Moving forward, there is a need to identify surveying strategies that provide the rigor, reliability and validity of the CASPER method while enabling a focus on equity in a less resource-intensive manner. With an increased focus on racial equity and environmental justice at the national level [54], we recommend CDC or other implementers assess potential modifications to CASPER to consider equity. Such modifications might include adaptations to oversample specific populations and facilitate the analysis of health disparities. For example, it is possible to select additional clusters and use a three-stage cluster sampling approach to facilitate intra-sample comparisons (e.g., between neighborhoods). In our case, we could have stratified first by neighborhood, and then selected 20 (for example) clusters from each neighborhood, potentially allowing for cross-neighborhood comparisons. However, the relatively small size of each of the neighborhoods of interest would have introduced challenges in obtaining the requisite number of households. CASPERs, including a recent example from 2020 in Chelan-Douglas County in Washington, USA, [55] have also been done using only census blocks that met certain racial/ethnic, socio-economic, or other criteria based on census data (e.g., only including blocks that had 50% of households with at least one household member identifying as a particular race/ethnicity) to focus on the population of interest. Similar recent innovative CASPER adaptations, including a study in Alabama that modified CASPER to assess health disparities between Hispanic and non-Hispanic White populations [56], should be collated, reviewed, and evaluated to assess replicability and validity of the approaches.

In place of CASPER, however, researchers can consider other surveying strategies (some of which are less labor- and resource-intensive) that may provide adequate and appropriate data to meet a project team's needs. Specifically, SASPER evaluation interviewees discussed that, following a flooding incident in South Park, community service providers that serve specific racial/ethnic groups conducted door-knocking multiple times a day to every household in the impacted area. This micro-targeted and comprehensive outreach enabled bi-directional, language-appropriate communication and support to community members in the immediate post-disaster phase; however, such approaches may not be scalable to disasters with broader impact areas. Fig 3 discusses tradeoffs of the CASPER approach, and options to consider based on partner priorities and project goals.

## Limitations

Limitations of the SASPER survey include constraints of the CASPER method and limitations related to the SASPER implementation of the approach. Specifically, a CASPER is intended to provide data representative of the community in which it is conducted: the data are not generalizable beyond Seattle's Duwamish Valley [25]. Furthermore, the data are cross-sectional: it captures the perspectives of community members during a specific moment in time. Had the SASPER been completed at a different time in the year, for example, following a serious flooding incident in December 2022, the results may have been different as community members' priorities and perspectives shifted. Regarding the implementation of the approach, the team

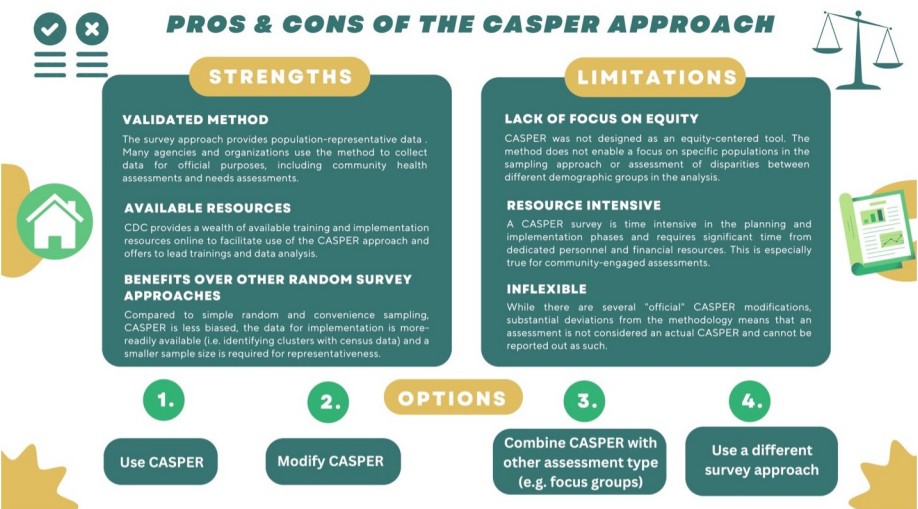

**Fig 3. The key strengths and limitations of the CASPER approach, as identified through the SASPER evaluation process and recommended options for project teams following consideration of the method tradeoffs.**

did not complete the minimum number of surveys (80%) for which the CASPER approach has been validated, resulting in an unacceptably low number of completed interviews, too low to accurately represent the sampling frame. Specifically, a minimum 168 surveys (with a goal of 210) was required, while the SASPER survey collected 162 surveys within the sampling frame [25]. Additionally, the SASPER included several modifications outside of the CASPER approach, such as the inclusion of surveys from unhoused individuals and use of online surveys. However, our study's inclusion of unhoused individuals, a population that is both disproportionately at-risk for experiencing negative impacts from environmental hazards and usually excluded from CASPER surveys and other types of community needs assessments, allowed our data to better represent community concerns and priorities and should be viewed as a strength.

There are also several limitations to the evaluation of the SASPER project. Respondents to the day-of feedback surveys had, overall, strong positive responses. While this agreement may have been legitimate, there also may have been a demand effect in which respondents felt compelled to be more favorable about the effort [57]. To mitigate this, surveys were anonymous to allow for more-candid feedback. Interviews with key team-members were conducted several months following the surveying effort. While this may have allowed partners more time to reflect on and fully-form their perspectives, the time lag could also have contributed to recall bias, limiting feedback specificity. Additionally, the data for the evaluation were only obtained from volunteers in the surveying and those deeply involved in the project. As such, the impacts of the effort on community members and their perspectives on the project are not in the scope of the evaluation. Finally, there are benefits and limitations to this being an internal evaluation. While there may be less perceived objectivity with an internal evaluation, the approach also enables a deeper understanding of and familiarity with the subject matter than with an external evaluation [58].

## Conclusions

As communities, and particularly those disproportionately affected by environmental injustice, grapple with worsening impacts of climate change, it is increasingly important to develop

and test strategies for equitably identifying community-level priorities for resilience actions. In response, academic researchers, alongside community and government partners, co-produced the "SASPER" project to document household and community-level climate, health, and disaster priorities to build resilience in Seattle's Duwamish Valley communities. Using a modified approach to CDC's CASPER, the team identified that Duwamish Valley community members are very concerned about climate and environmental hazards, the community has a strong level of connectedness, and community members want to be involved in resilience building efforts. In addition, an internal, mixed methods, culturally responsive evaluation combined survey data from SASPER volunteers with in-depth interviews with core SASPER team-members to assess the feasibility, appropriateness, and effectiveness of the project. Findings suggest the method is limited in its ability to highlight and assess the concerns and priorities of populations that have been marginalized. Specifically, CASPER may be considered as an equality-oriented survey strategy (i.e., treating everyone the same), instead of an equity-oriented strategy, which would recognize the differences faced by certain subpopulations of a community. As CASPER is frequently used to make important decisions in disaster contexts and in public health planning, it is critical that researchers evaluate the equity implications of the method. As such, we recommend that future projects that use CASPER conduct an evaluation of their project, including equity considerations, and that future research identify and evaluate ways to center equity in the approach.

## Supporting information

**S1 Checklist. Human participants research checklist.**
(DOCX)

## Acknowledgments

We would like to thank the Duwamish Valley community who completed the survey, the Duwamish Valley Youth Corps, University of Washington, and Public Health Reserve Corps volunteers who conducted the surveys and made this project possible, Amy Helene Schnall from the Centers for Disease Control and Prevention, as well as Graciela Flores, Ashley Moore, and Lisa Hayward Watts from the University of Washington for their contributions.

## Author Contributions

**Conceptualization:** Samantha Lovell, Jamie Vickery, Paulina López, Alberto J. Rodríguez, B. J. Cummings, Jeffrey Berman, Ann Bostrom, Tania Busch Isaksen, Cat Hartwell, Pamela Kohler, C. Bradley Kramer, Resham Patel, Amy Helene Schnall, Nicole A. Errett.

**Data curation:** Samantha Lovell, Jamie Vickery, Kathleen Moloney, Jeffrey Berman, Nicole A. Errett.

**Formal analysis:** Samantha Lovell, Kathleen Moloney, Ann Bostrom, Amy Helene Schnall, Nicole A. Errett.

**Funding acquisition:** Jeffrey Berman, Ann Bostrom, Nicole A. Errett.

**Investigation:** Samantha Lovell, Jamie Vickery, Paulina López, Alberto J. Rodríguez, B. J. Cummings, Kathleen Moloney, Ann Bostrom, Tania Busch Isaksen, Cat Hartwell, Pamela Kohler, C. Bradley Kramer, Resham Patel, Amy Helene Schnall, Mary Hannah Smith, Nicole A. Errett.

**Methodology:** Jamie Vickery, Alberto J. Rodríguez, B. J. Cummings, Jeffrey Berman, Ann Bostrom, Tania Busch Isaksen, Erika Estrada, Cat Hartwell, Pamela Kohler, C. Bradley Kramer, Resham Patel, Amy Helene Schnall, Mary Hannah Smith, Nicole A. Errett.

**Project administration:** Samantha Lovell, Jamie Vickery, Paulina López, Alberto J. Rodríguez, B. J. Cummings, Kathleen Moloney, Erika Estrada, Cat Hartwell, C. Bradley Kramer, Resham Patel, Mary Hannah Smith, Nicole A. Errett.

**Resources:** Jamie Vickery, Paulina López, Alberto J. Rodríguez, B. J. Cummings, Jeffrey Berman, Ann Bostrom, Erika Estrada, C. Bradley Kramer, Resham Patel, Amy Helene Schnall, Nicole A. Errett.

**Software:** Jamie Vickery, Jeffrey Berman.

**Supervision:** Samantha Lovell, Jamie Vickery, Paulina López, Alberto J. Rodríguez, B. J. Cummings, Kathleen Moloney, Cat Hartwell, Amy Helene Schnall, Mary Hannah Smith, Nicole A. Errett.

**Validation:** Kathleen Moloney, Tania Busch Isaksen, Amy Helene Schnall, Nicole A. Errett.

**Visualization:** Samantha Lovell.

**Writing – original draft:** Samantha Lovell, Nicole A. Errett.

**Writing – review & editing:** Samantha Lovell, Jamie Vickery, Paulina López, Alberto J. Rodríguez, B. J. Cummings, Kathleen Moloney, Jeffrey Berman, Ann Bostrom, Tania Busch Isaksen, Erika Estrada, Cat Hartwell, Pamela Kohler, C. Bradley Kramer, Resham Patel, Amy Helene Schnall, Mary Hannah Smith, Nicole A. Errett.

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
