## [Decision Letter · Decision Letter 0]

13 Dec 2023

PONE-D-23-32934A community-engaged approach to assess climate resilience and disaster priorities: Survey results and evaluationPLOS ONE

Dear Dr. Errett,

Thank you for submitting your manuscript to PLOS ONE. After careful consideration, we feel that it has merit but does not fully meet PLOS ONE’s publication criteria as it currently stands. Therefore, we invite you to submit a revised version of the manuscript that addresses the points raised during the review process.

We look forward to receiving your revised manuscript.

Kind regards,

Olushayo Oluseun Olu

Academic Editor

PLOS ONE

Reviewers' comments:

Reviewer's Responses to Questions

**Comments to the Author**

1. Is the manuscript technically sound, and do the data support the conclusions?

Reviewer #1: Yes

Reviewer #2: Yes

2. Has the statistical analysis been performed appropriately and rigorously? 

Reviewer #1: Yes

Reviewer #2: N/A

3. Have the authors made all data underlying the findings in their manuscript fully available?

Reviewer #1: Yes

Reviewer #2: Yes

4. Is the manuscript presented in an intelligible fashion and written in standard English?

Reviewer #1: Yes

Reviewer #2: Yes

5. Review Comments to the Author

Reviewer #1: This study used a modified CDC CASPER approach and conducted evaluations of the approaches. The results, discussion, and conclusion section of the manuscript are well written, and the findings are important to academics and practitioners. The introduction needs to be improved as the rationale of the study needs to be expanded. The structure of the introduction is also disjointed. I would also add more details on the results section about the resources needed for the CASPER/SASPER.

Introduction:

• Remove the comma that comes after Participatory

“Participatory, community-centered research can help demonstrate the disproportionate impacts of climate change on low-income populations, people of color, older adults, people living unhoused, and other groups that have been marginalized.”

• A paragraph introducing CASPER is needed in the introduction. In this paragraph, I would also include the rationale for using CASPER as a starting point to assess climate resilience. I would also expand on the rationale of using the SASPER.

• The introduction could be improved significantly to frame the objectives of the manuscript better, and most importantly, to provide more context about the rationale of the study. I would recommend providing more depth to the rationale of the research. The authors identify some of the advantages of the co-production of climate knowledge and bottom-up approaches in the energy sector. I would recommend expanding on this to provide a more robust rationale for the study.

• The transition to the “Setting” section seems disjointed from the introduction. I would recommend embedding the setting of the study early on in the introduction as a way to improve the rationale of the study.

• I would also provide information on the possible scalability/replicability (or not) of the tool in other contexts.

Methodology

• The first sentence of the overview is a bit hard to read. I suggest to break this in two sentences. One with the collaborating institutions, and another with the aims.

• In the sentence below, from the community partner section, the word “level” seems to be misplaced. Also, the citation is after the period and should be moved.

“To promote research reciprocity, improve transparency, level real or perceived power differentials, and build trust, we grounded our work in a written community partnership agreement.[27]”

• I would include the nine languages the instruments were translated to. Also, I suggest including a statement about how translations were made to ensure that the meaning of the surveys was not lost (e.g., translated by members of the study team, or official translating services).

• I would provide the number of surveys done with SASPER volunteers in the main text and not only in the figure.

•

Results

• I would also suggest including how many surveys were completed in each language (English=X, Spanish=Y, etc.).

• There are several mentions of the approach being resource and labor-intensive. It would benefit readers and those who wish to implement the tool, if the authors could provide more detail on the required resources.

• I would reword the volunteer experience section of the results to more accurately represent the findings. The study did not use measures to assess whether volunteers developed skills or achieved any learning outcomes. If this results are based on the interviews, phrases such as the one below should be modified to say that participants perceived an improvement in those skills.

“Youth volunteers developed new interpersonal and public speaking skills, learned about public health and research methods”

Reviewer #2: The paper is well written and demonstrates an objective and comprehensive evaluation of the process undertaken. It is technically sound having applied sound qualitative analysis on the users descriptive experience of the application of an adopted form of the research tool. From a readers perspective, the paper content (particularly the description of the findings and discussion section) is predominantly focused on this in-depth evaluation of the process rather than the results of the survey (i.e. the qualitative aspect). A suggestion to the authors would be to consider changing the title to reflect that. For example, Evaluation of a community assessment tool used to assess……..and survey results. A summary of recommendations in bullet form might also be useful.

6. PLOS authors have the option to publish the peer review history of their article (what does this mean?). If published, this will include your full peer review and any attached files.

Reviewer #1: No

Reviewer #2: No

---

## [Author Response · Author response to Decision Letter 0]

28 Feb 2024

We very much appreciate the thoughtful critique of our manuscript previously titled, “A community-engaged approach to assess climate resilience and disaster priorities: Survey results and evaluation”, which we have now titled, “Evaluating an equity-focused approach to assess climate resilience and disaster priorities through a community survey” in response to Reviewer comments. Below we respond to each of the Reviewers’ comments raised in the December 13th, 2023 email correspondence. Reviewer comments are provided in italics and are followed by itemized responses. Please note every response is linked to a page and line demarcation in the revised manuscript submission.

Journal Requirements: 

Thank you for providing these references. We have confirmed that our revised manuscript complies with PLOS ONE’s style requirements, as outlined in the documents provided.

We have reviewed our reference list for completeness and accuracy. As part of the revisions requested by the Reviewers, we have added the following additional references:

1. Cascadia Consulting Group. Seattle Climate Vulnerability Assessment. Seattle Office of Planning & Community Development; 2023 Jun. Available: https://seattle.gov/documents/departments/opcd/seattleplan/seattleclimatevulnerabilityassessmentjuly2023.pdf

2. Murti M, Bayleyegn T, Stanbury M, Flanders WD, Yard E, Nyaku M, et al. Household emergency preparedness by housing type from a community assessment for public health emergency response (CASPER), Michigan. Disaster Med Public Health Prep. 2014;8: 12–19.

3. Ferguson RW, Kiernan S, Spannhake EW, Schwartz B. Evaluating Perceived Emergency Preparedness and Household Preparedness Behaviors: Results from a CASPER Survey in Fairfax, Virginia. Disaster Med Public Health Prep. 2020;14: 222–228.

Reviewers' comments:

Reviewer #1: 

1. This study used a modified CDC CASPER approach and conducted evaluations of the approaches. The results, discussion, and conclusion section of the manuscript are well written, and the findings are important to academics and practitioners. The introduction needs to be improved as the rationale of the study needs to be expanded. The structure of the introduction is also disjointed. I would also add more details on the results section about the resources needed for the CASPER/SASPER.

We thank Reviewer #1 for their thorough and overall positive review of the manuscript, and agree that the findings presented in the manuscript are of interest to both academics and practitioners. We appreciate Reviewer #1’s suggestions to improve the Introduction section of the manuscript, and have added additional text to this section to describe the study rationale and setting (see items #4 and #5 below) and to provide an overview of the CASPER process (see item #3). We have also added a description of the resources required to conduct the SASPER to the Discussion section (see item #12 below). 

Introduction:

2. Remove the comma that comes after Participatory

“Participatory, community-centered research can help demonstrate the disproportionate impacts of climate change on low-income populations, people of color, older adults, people living unhoused, and other groups that have been marginalized.”

We agree with Reviewer #1’s suggestion to remove the comma in the above sentence, and have done so in the revised manuscript (page 5, line 116). 

3. A paragraph introducing CASPER is needed in the introduction. In this paragraph, I would also include the rationale for using CASPER as a starting point to assess climate resilience. I would also expand on the rationale of using the SASPER.

We thank Reviewer #1 for their suggestion to include an overview of the CASPER method in the Introduction. We have added a paragraph that provides an overview of CASPER and its previous use in pre-disaster contexts to this section, to clarify the rationale for choosing this method for our project. We have also clarified in this paragraph that the SASPER approach employed the CASPER method, with several additional steps to ensure the project centered community voice and equity (page 6, lines 142-157). 

4. The introduction could be improved significantly to frame the objectives of the manuscript better, and most importantly, to provide more context about the rationale of the study. I would recommend providing more depth to the rationale of the research. The authors identify some of the advantages of the co-production of climate knowledge and bottom-up approaches in the energy sector. I would recommend expanding on this to provide a more robust rationale for the study.

We appreciate Reviewer #1’s comment that we could improve our framing of the study’s objectives and rationale in the Introduction section of the manuscript. To better clarify the objectives and rationale, we have added text describing the pre-existing network of community partnerships in the Duwamish Valley to support successful community-engaged research, the purpose of newly established Duwamish Valley Resilience District, and that our study intends to complement and inform the efforts of the Duwamish Valley Resilience District (pages 5-6, lines 131-145). We have also included text describing the limited prior evaluations of the CASPER method, and the lack of evidence exploring how community-engaged approaches can foster community partnerships and advance research on climate health impacts, as rationale for conducting an evaluation of our chosen survey approach (pages 6-7, lines 159-176). 

5. The transition to the “Setting” section seems disjointed from the introduction. I would recommend embedding the setting of the study early on in the introduction as a way to improve the rationale of the study.

We agree with Reviewer #1 that the details originally provided in the Setting section of the manuscript are important to help readers understand the rationale for our study. We have now embedded this section at the beginning of the Introduction section (pages 3-4, lines 73-99). 

6. I would also provide information on the possible scalability/replicability (or not) of the tool in other contexts.

We appreciate Reviewer #1’s suggestion to add commentary on the replicability and scalability of the CASPER approach. We have added text in the Introduction that clarifies that CASPER has been replicated in other contexts (page 6, lines 149-155). We have also added comments in the Discussion section that discuss how the resource-intensiveness of our adapted CASPER approach may limit its scalability and replicability as a tool for conducting community-engaged research (page 34, lines 943-946). 

Methodology

7. The first sentence of the overview is a bit hard to read. I suggest to break this in two sentences. One with the collaborating institutions, and another with the aims.

We appreciate Reviewer #1’s suggestion to break apart this sentence, and agree that separating the collaborating institutions and the aims into separate sentences improves clarity. The sentences now read, “Building on long-standing community partnerships, the SASPER project is a collaborative effort between the University of Washington, the City of Seattle, the Duwamish River Community Coalition and the organization’s Duwamish Valley Youth Corps, the Washington State Department of Health, and Public Health–Seattle & King County. The project aims to assess household- and community-level climate change and health impacts, access to and needs for information and resources to promote resilience, and pathways for community input into ongoing climate change adaptation planning. In addition, it aims to evaluate the CDC’s CASPER method, as well as the modifications to this method that sought to center equity and community in the process.” (pages 7-8, lines 190-292). 

8. In the sentence below, from the community partner section, the word “level” seems to be misplaced. Also, the citation is after the period and should be moved.

“To promote research reciprocity, improve transparency, level real or perceived power differentials, and build trust, we grounded our work in a written community partnership agreement.[27]”

We agree with Reviewer #1 that the meaning of the word “level” is unclear in this sentence. We have replaced it with the word “balance”, which more clearly communicates the intended meaning. We also thank Reviewer #1 for pointing out that our citation was misplaced, and have moved it before the period (page 8, line 297)

9. I would include the nine languages the instruments were translated to. Also, I suggest including a statement about how translations were made to ensure that the meaning of the surveys was not lost (e.g., translated by members of the study team, or official translating services).

We agree with Reviewer #1’s comment that we should list the languages that the survey instrument was translated into, and have now listed all the languages in the text . We have also clarified the wording slightly, to make it clear that the nine languages the survey was offered in includes English (page 10, lines 430-432). We have also added text to clarify that a professional translation service was used to ensure the accuracy and clarity of translated surveys (page 10, line 430). 

10. I would provide the number of surveys done with SASPER volunteers in the main text and not only in the figure.

We thank Reviewer #1 for pointing out that we should also include the number of volunteer feedback surveys we collected in the manuscript text. We have now provided this information at the beginning of the Feedback Survey section under Results (page 21, lines 629).

Results

11. I would also suggest including how many surveys were completed in each language (English=X, Spanish=Y, etc.).

We agree with Reviewer #1 that we should provide a breakdown of the languages in which the paper surveys were completed. We have added text reporting this breakdown in the SASPER Survey section under Results (page 16, lines 576-578). 

12. There are several mentions of the approach being resource and labor-intensive. It would benefit readers and those who wish to implement the tool, if the authors could provide more detail on the required resources.

We thank Reviewer #1 for their suggestion to include more details about the resources required to design and implement the SASPER project, and agree that this information will benefit readers. We have added several sentences to the Discussion section that describe the grant funds, in-kind donations, and volunteer hours that supported the project (pages 34, lines 923-935). 

13. I would reword the volunteer experience section of the results to more accurately represent the findings. The study did not use measures to assess whether volunteers developed skills or achieved any learning outcomes. If this results are based on the interviews, phrases such as the one below should be modified to say that participants perceived an improvement in those skills.

“Youth volunteers developed new interpersonal and public speaking skills, learned about public health and research methods”

We appreciate Reviewer #1’s comment that the sentence above should be modified to clarify that we are reporting the results of evaluation interviews. We have added text to the beginning of this sentence to clarify this; the sentence now reads, “Interviewees with insight into the experiences of the youth volunteers described how they had developed new interpersonal and public speaking skills, learned about public health and research methods, and built confidence by being part of the broader project effort related to community priorities” (page 27, lines 753-756). 

Reviewer #2: 

1. The paper is well written and demonstrates an objective and comprehensive evaluation of the process undertaken. It is technically sound having applied sound qualitative analysis on the users descriptive experience of the application of an adopted form of the research tool. From a readers perspective, the paper content (particularly the description of the findings and discussion section) is predominantly focused on this in-depth evaluation of the process rather than the results of the survey (i.e. the qualitative aspect). A suggestion to the authors would be to consider changing the title to reflect that. For example, Evaluation of a community assessment tool used to assess……..and survey results. 

We greatly appreciate Reviewer #2’s thoughtful review of and positive feedback on our manuscript. We agree that the original title did not align with the manuscript’s predominant focus on the evaluation. We have updated the title to “Evaluating an equity-focused approach to assess climate resilience and disaster priorities through a community survey” in order to better reflect the manuscript’s content. 

2. A summary of recommendations in bullet form might also be useful.

We thank Reviewer #2 for their comment. We have included a summary of the specific process recommendations provided by interviewees who participated in the evaluation in Table 2 (pages 24-25). We welcome additional suggestions from Reviewer #2 about how to present the recommendations in the format that is most useful for readers.

---

## [Editor Report · Decision Letter 1]

28 Mar 2024

Evaluating an equity-focused approach to assess climate resilience and disaster priorities through a community survey

PONE-D-23-32934R1

Dear Dr. Errett,

We’re pleased to inform you that your manuscript has been judged scientifically suitable for publication and will be formally accepted for publication once it meets all outstanding technical requirements.

Kind regards,

Olushayo Oluseun Olu

Academic Editor

PLOS ONE